# Experimental Analysis of the Space Ratio Influence on the Excitation Frequencies of One and Two Cylinders Free to Vibrate in Tandem Arrangement

**Roberta Fátima Neumeister, Adriane Prisco Petry and Sergio Viçosa Möller ***

Graduate Program in Mechanical Engineering PROMEC, Federal University Rio Grande do Sul UFRGS, Porto Alegre 91501-970, RS, Brazil
* Correspondence: svmoller@ufrgs.br

**Abstract:** The present study aims to investigate the dominant frequency ranges of a cylinder free to vibrate transversally to the flow positioned in the first, the second or in both positions of the tandem assembly for L/D = 1.26, 1.4, 1.6, and 3.52 with the increase in the flow velocity. Accelerometers and hot wire anemometers were the experimental tools applied in this study. The range of study encompassed the reduced velocity with values from 6 to 72 and Reynolds number from $7.1 \times 10^3$ to $2.4 \times 10^4$. Fourier transform, continuous wavelet transform, magnitude-square coherence, and wavelet coherence were applied to analyze the cylinder acceleration results for all L/D and wake velocity values studied. The results show that the amplitudes of vibration are below 1.5% of the diameter for all the cases, except for the lower L/D, where the amplitude increases. The first cylinder free to vibrate presents the highest amplitudes observed. Fourier and continuous wavelet analysis showed high energy associated with the two natural frequencies of the system and a third frequency, which may be associated with the flow excitation. In the second cylinder free to vibrate, energy spreads across the monitored spectrum, justifying the smaller amplitudes but the energy level increases with increasing L/D and may be associated with wake-induced vibration. The cases with both cylinders free to vibrate show that the relation between the assembly parameters of each cylinder is relevant to the vibration response and the excitation frequency range. The results showed that even with a clear excitation in a higher frequency, the main energy in the vibration signals is in the natural frequency range.

**Keywords:** flow-induced vibration; cylinders; tandem arrangement; wavelets; acceleration

## 1. Introduction

The flow-over cylinder arrangement is found in many engineering applications, such as chimneys, transmission lines, and heat exchangers, as well as in ocean engineering applications, e.g., in offshore structures and oil platforms, risers, and pipelines. This study, using pairs of cylinders, can help to understand phenomena in complex arrangements, such as tube banks, due to the close space ratio between the cylinders causing many mechanisms of interaction to be present. Studies using pairs of cylinders can help to understand the interaction between cylinders side-by-side or in tandem and the results can be extrapolated to the models for complex assemblies.

The analysis of flow over fixed tandem cylinders has been studied in the past decades focusing on understanding the wake behavior. In 1981, Igarashi [1] presented a study with cylinders in tandem observing the wake patterns for different L/D ratios and Reynolds numbers. An in-depth review on the flow, pressure distribution, Strouhal number, drag, and lift coefficients for fixed cylinders in tandem, considering the influence of the Reynolds number and the space ratio was presented in 2003 by Zdravkovich [2]. The results were classified according to the patterns observed. Studies with tandem cylinders to

characterize the flow patterns and detail drag and lift forces were presented by Alam et al. [3] and Sumner [4].

Pairs of cylinders in the tandem configuration are characterized by one cylinder positioned after the other, as presented in Figure 1. The space between the cylinders, L, is a dominant variable in the analysis and is usually treated as the space ratio L/D. In the tandem arrangement, the flow characteristics, the diameter of cylinders, and the distance between them are relevant to the analysis.

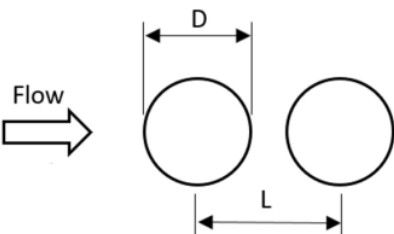

**Figure 1.** Flow over tandem cylinders.

Despite the interest of many researchers in the flow on pairs of fixed cylinders in the past few years, the lack of information about cylinders free to vibrate still remains. A revision concerning free to vibrate pairs of cylinders was presented by Païdoussis et al. [5], including models and experimental results, showing the main mechanisms associated with pairs of cylinder vibrations and the areas that need to be explored. The main models considered static instability, but the dynamic instability occurrence remains with the initial results. In the tandem configuration, the dominant mechanism changes as the spacing ratio increases.

Studies concentrate on high space ratios to investigate the wake-induced vibration (WIV) related to the wake from the first cylinder. The studies of WIV generally use L/D values greater than 5 to develop the wake between both cylinders. Coherent vortices interfering with the downstream cylinder induce fluctuations in the fluid forces not synchronized with the motion, as shown in [6]. In the case of two tandem cylinders free to oscillate, both in crossflow and stream-wise directions, it was observed that the first and the second cylinders oscillated at different frequencies. These results contrast with those obtained for two fixed cylinders in tandem, where the wakes of both cylinders oscillated at the same frequency [7].

In a numerical study, the local unsteady characteristics of the transverse wake-induced vibration was investigated [8]. The upstream cylinder was fixed while the downstream one was free to vibrate perpendicularly to the freestream flow in a three-dimensional simulation. The results showed that the wake-excitation mechanism is sustained by the interaction of upstream vortical wake with the freely downstream cylinder. The large displacement of the downstream cylinder is due to the appearance of a low-frequency component in the transverse load, which is closer to the natural frequency of the downstream cylinder. It was shown that the boundary layer movement on the downstream cylinder plays a major role in sustaining the low-frequency component in the transverse load.

For applications with a close space ratio, additional flow-induced vibration (FIV) mechanisms should be investigated. Four vibration regimes based on the characteristics and generation mechanisms of the cylinder galloping vibrations were observed by Qin et al. [9]. The initial states of a cylinder have a pronounced impact on the vibration of the other. Alternating reattachment, detachment, rolling up, and shedding of the upper and lower gap shear layers all contribute to the vibrations. The authors of [10] showed that the FIV characteristics of the upstream cylinder are similar to a single-cylinder for L/D over 2.5 when the reduced velocity does not exceed 12.0. The FIV response of the upstream

cylinder in the two tandem cylinder system with T/D = 1.57 is significantly different from that of the isolated cylinder. The vibration amplitude and the fluid force on the downstream cylinder are influenced by the upstream cylinder for the spacing range tested.

In a study on the flow-induced vibration of an elastically mounted circular cylinder [11], the results showed that the downstream cylinder exhibits three regimes of vibration responses: pure vortex resonance, separated vortex resonance, wake-induced galloping, and combined vortex resonance with wake-induced galloping. The L/D for the occurrence of each regime will be changed depending on the mass-damping parameter and initial conditions. An experimental study with a flexible cylinder placed in the wake of a stationary rigid cylinder in tandem configuration with L/D ratios ranging from 3 to 9 was presented in Ref. [12]. The flexible cylinder was free to vibrate in transversal and in-line directions. The dynamic response observed in this study resulted from two combined phenomena of vortex-induced vibration and wake-interference galloping.

The use of one cylinder free to vibrate and one fixed on a tandem configuration can represent situations in engineering devices where there is a high difference in the stiffness between the cylinders: one presents a low natural frequency and the other present a high natural frequency. In the previous studies of our research group, it was observed that in close spaced cylinder arrangements, the tandem configuration has a clear impact on the amplitudes of vibration.

In the study presented in Ref. [13], the tandem configuration with L/D = 1.26 with the first cylinder free to vibrate presented the higher amplitudes of vibration. While the tandem case with the second cylinder free to vibrate presented the equivalent response to a single cylinder. It was observed that the presence of the cylinder in the wake of the cylinder free to vibrate amplified the amplitude response at high reduced velocities. In the cases of the first or the second cylinder free to vibrate, the amplitudes increase for a blockage ratio of 13% while, for a blockage ratio of 26.8%, a jump in the vibration amplitude happens for high flow velocities. In the cases of both cylinders free to vibrate, the increase on the amplitude is higher for the first cylinder but the amplitude jump occurs for the second cylinder. The results for increasing blockage ratios include the energy distribution in different frequency bands and the change of the excitation mode from the second to the first mode.

In [14], the analysis of FIV in tube banks with square arrangement indicates that the sudden increase in acceleration is linked to an excitation close to the natural frequency range. The increase in the acceleration alters the main frequencies of the flow, leading to an interaction between vibration and flow velocity, which are mutually reinforcing. The initial excitation may be related to shear layer oscillations, presented in visualizations in the literature, which, for small space ratios, can have a significant influence.

FIV studies on tandem cylinders can be simplified as a fixed cylinder and a cylinder free to vibrate. In the literature, some studies showed that the first cylinder influences the second cylinder vibration pattern, but most are for low Reynolds numbers and reduced velocity around 5. The studies considered the second cylinder free to vibrate or both cylinders free to vibrate, but few of them consider the case of the first cylinder free to vibrate. Close space ratio, high reduced velocity, and different natural frequency conditions of the cylinders are not usually found in the literature, but they are conditions identified in some applications, such as tube banks. The study with tandem cylinders in these configurations can help to identify the dominant mechanisms of excitation in a simplified structure and help in the interpretation of the main mechanisms in more complex flows.

Based on the available information, the present study aims to investigate the dominant frequency ranges of a cylinder free to vibrate in a cylinder set in a tandem arrangement with the space ratio varying from 1.26 to 3.52 using available tools to investigate the frequency domain and the time–frequency domain. The cylinder free to vibrate is positioned firstly upstream of the second cylinder and then downstream to identify the influence of the space ratio on the frequency excitation observed. An analysis with both cylinders free to vibrate is performed for the space ratio of 1.26.

## 2. Materials and Methods

The experimental analysis was performed in one of the aerodynamic channels of the Fluid Mechanics Laboratory—LMF of UFRGS. The aerodynamic channel employed has a rectangular test section of 0.146 m in height and a width of 0.193 m with acrylic walls. A centrifugal blower of 0.75 kW impels the air through a diffuser, two honeycombs, and two screens to reduce the turbulence intensity to less than 1% of the free stream velocity in the test section. The reference velocity is measured with a Pitot tube positioned upstream of the test section in the non-perturbed flow. The Reynolds number, $Re = VD/v$, computed with the tube diameter and the main flow velocity, ranged between $7.1 \times 10^3$ and $2.4 \times 10^4$. Details about the aerodynamic channel are presented in Figure 2a.

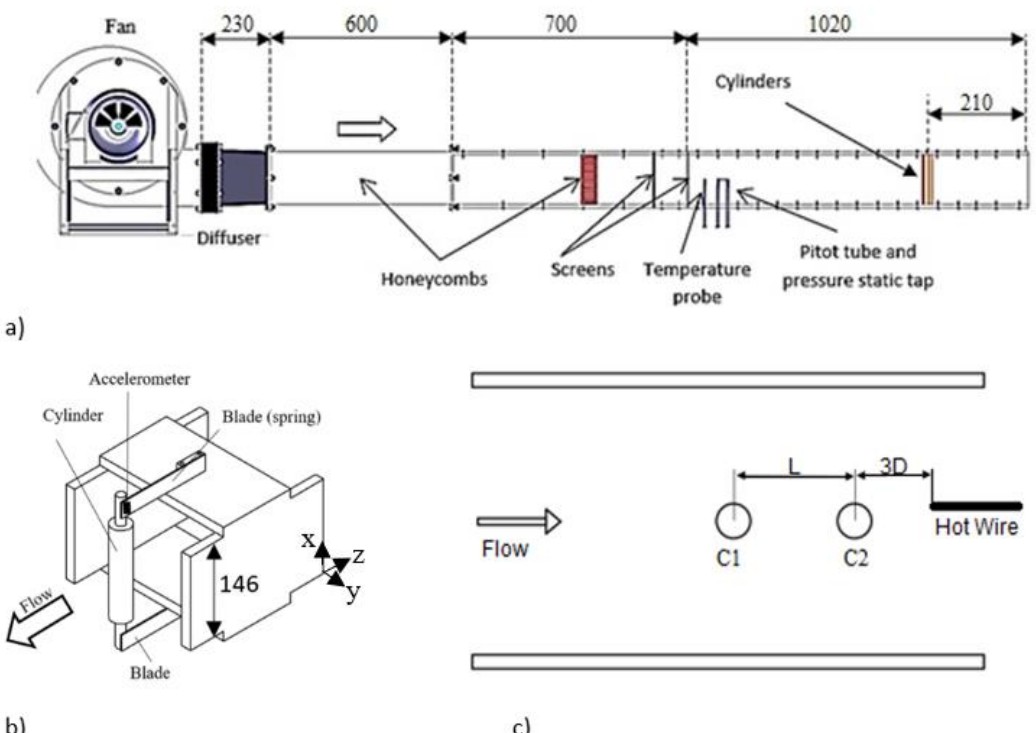

**Figure 2.** (**a**) Aerodynamic channel (dimensions in millimeters), (**b**) schematic detail of the assembly of the cylinder free to vibrate, and (**c**) schematic top view of the test section with the configuration of the first cylinder and second cylinder and hot wire position.

The test section is assembled with smooth cylinders of 25 mm diameter. One of the cylinders is attached to two stainless steel blades that allow vibrating in the transversal direction, as presented in Figure 2b. The tandem arrangements have one or both cylinders free to vibrate. In Figure 2c, a schematic of the assembly top view is presented.

In the studied cases, the cylinders are in tandem arrangement with one or both cylinders free to vibrate. The description of the cases with FV means that the first cylinder is free to vibrate and the description with SV means that the second cylinder is free to vibrate. The cases are noted with FV or SV followed by the SV-ratio. The cases with both cylinders free to vibrate are described with the BV prefix and the complement of *C1* and *C2* to indicate which cylinder is been monitored.

The acceleration in the y-direction, Figure 2b, of the cylinder free to vibrate is the monitored variable. The cylinder displacement was obtained from the acceleration measurement, using an ADXL335 accelerometer with a range of ±3 g and a resonant frequency of 5.5 kHz. The error associated with the acceleration signals is about ±7 %. The velocity in the wake is measured concomitant with the acceleration signal. Data acquisition was

performed using an A/D board 16 bits NI USB-9162 with a sampling frequency of 1 kHz and an anti-aliasing low pass filter at 0.3 kHz.

Table 1 presents all the investigated cases.

The mass-damping parameters were constant for the Cases 1 to 9 with changes in the space ratio. The mass ratio is the relation between the cylinder mass and the mass of fluid displaced during the movement. The damping ratio is the relation between the energy dissipated in a cycle and the total structure energy in vibration. The test section assembly is equivalent to the one presented in [13]. The parameter of the mass ratio changed with the cylinder mass used in the assembly. The damping ratio changed with the length and thickness of the blades applied.

**Table 1.** Tested cases parameters.

| Case | | m* | $\zeta$ | L/D | $f_{n1}$ | $f_{n2}$ | Free to Vibrate |
|------|------|------|------|------|------|------|------|
| Case 01 | Single Cylinder | 608 | 0.03 | - | 7.8 | 21.5 | SC |
| Case 02 | FV-1.26 | 608 | 0.03 | 1.26 | 7.8 | 21.5 | FV |
| Case 03 | SV-1.26 | 608 | 0.03 | 1.26 | 7.8 | 21.5 | SV |
| Case 04 | FV-1.4 | 608 | 0.03 | 1.4 | 7.8 | 21.5 | FV |
| Case 05 | SV-1.4 | 608 | 0.03 | 1.4 | 7.8 | 21.5 | SV |
| Case 06 | FV-1.6 | 608 | 0.03 | 1.6 | 7.8 | 21.5 | FV |
| Case 07 | SV-1.6 | 608 | 0.03 | 1.6 | 7.8 | 21.5 | SV |
| Case 08 | FV-3.52 | 608 | 0.03 | 3.52 | 7.8 | 21.5 | FV |
| Case 09 | SV-3.52 | 608 | 0.03 | 3.52 | 7.8 | 21.5 | SV |
| Case 10 | FV-1.26-1 | 539 | 0.004 | 1.26 | 8.8 | 23.4 | FV |
| Case 11 | SV-1.26-1 | 539 | 0.004 | 1.26 | 8.8 | 23.4 | SV |
| Case 12-C1 | BV-C1-1.26-Config1 | 502 | 0.003 | 1.26 | 13.7 | 36.2 | BV-C1 |
| Case 12-C2 | BV-C2-1.26-Config1 | 547 | 0.006 | 1.26 | 24.4 | 71 | BV-C2 |
| Case 13-C1 | BV-C1-1.26-Config2 | 502 | 0.002 | 1.26 | 29.3 | 58.6 | BV-C1 |
| Case 13-C2 | BV-C2-1.26-Config2 | 547 | 0.009 | 1.26 | 10.7 | 30.3 | BV-C2 |
| Case 14-C1 | BV-C1-1.26-Config3 | 502 | 0.008 | 1.26 | 10.8 | 22.5 | BV-C1 |
| Case 14-C2 | BV-C2-1.26-Config3 | 547 | 0.007 | 1.26 | 10.7 | 31.2 | BV-C2 |

The L/D and which cylinder was free to vibrate were the influence variables monitored. The value of the mass ratio was m* = 608 and the damping ratio, was $\zeta$ = 0.03. The natural frequencies are $f_{n1}$ = 7.8 Hz and $f_{n2}$ = 21.5 Hz. All the parameters were measured in the test section after the assembly of the system was free to vibrate in still air. The first frequency is related to movement of both blades to the same side and the second mode is related to the asymmetric movement of the blades. The information for each tested case is summarized in Table 1.

The influence of the first cylinder free to vibrate and the second cylinder free to vibrate on the wake velocity was investigated using L/D = 1.26, Cases 10 and 11. The assembly presented m* = 539, $\zeta$ = 0.004 and the natural frequencies were 8.8 Hz and 23.4 Hz. The velocity on the wake was measured with a hot wire probe positioned in the wake region of the second cylinder. In Figure 2c, the assembly top view with the position of the hot wire is presented. The hot wire probe is on a tangent to the cylinder and 75 mm from the center of the second cylinder.

The study with two cylinders free to vibrate and the space ratio of 1.26 were investigated for the case where both cylinders were in the same natural frequency range or with one cylinder with higher natural frequency than another (Cases 12 to 14). The assembly for the first cylinder free to vibrate presented m* = 502 and the natural frequencies ranged between were 10.8 Hz and 29.3 Hz, according to the tested case. The assembly for the second cylinder free to vibrate presented m* = 547 and the natural frequencies ranged between 10.7 Hz and 24.4 Hz for the tested cases.

The displacement amplitude of the cylinder free to vibrate, Y, is obtained from double integration of the acceleration signal, normalized with the diameter, D. The cylinder instrumented with the accelerometer had one degree of freedom and was assembled to vibrate transversally to the flow. The displacement was obtained by two successive integrations of the accelerometer signal using the trapezoid rule, with a double digital filter applied between the integrations to avoid the amplification of the low-frequency noise and maintain the phase of the signal. The results of amplitude are presented in this paper by the corresponding root mean square (RMS) values, using the same procedure as [13].

The reduced velocity, $U/Df_{n1}$, is based on the reference velocity obtained from the Pitot tube, the cylinder diameter, and the first natural frequency. The reduced velocity of the experiments presented values above 30 to explore the response over the resonance region. The uncertainties of the reduced velocities remain around 10.5%. The uncertainty associated with the frequency results is around ±9%.

The signal analysis was made in the frequency and time–frequency domains, employing power spectra and continuous wavelet spectra, respectively. In the present analysis, continuous wavelet transform (CWT) are performed applying Db20, while the Morlet functions more suitable to the coherence wavelet, due to its complex and nonorthogonal characteristics. The mathematical tools are described in [14] and [15].

The signal analysis was performed in the frequency domain using power spectrum and coherence function. The power spectrum of a given time function represents the signal energy level in the frequency domain considering the total time [16]. The coherence function between an input and output signal represents frequency ranges that have higher energy. The magnitude-squared coherence indicates the correspondence between two signals at each frequency, where the coherence function presents values between zero (no coherence) and one (100% coherence) [16].

The wavelet analysis allows a time–frequency evaluation due to the use of a scalable modulated window, shifted along with the signal. For every position, the spectrum is calculated, this process is repeated and the results in a time–frequency representation [17]. The wavelet function adopted in the analysis must represent well the characteristics of the raw signal. Studies carried on by [18] showed that the Db20 and Morlet functions are suitable for use with velocity signals. The use of wavelet coherence can indicate time and frequency correlation for two responses that, separately, do not indicate a clear relation. The wavelet coherence can be described as the non-dimensional response for the cross-wavelet, where the coherence between two signals is represented in time and frequency. The wavelet coherence presents values between zero and one, similarly to the Fourier coherence function.

Time series of all measurement results according to the freedom to vibrate condition were organized comprehensively in [19]. Data also include the case of the single cylinder.

## 3. Results and Discussions

The results are presented in four sections. The first section compares the non-dimensional displacement obtained from experiments with different space ratios L/D with the first or the second cylinder free to vibrate. The second section focus on the frequency ranges and patterns observed in the vibration of the cylinder with the change in the space ratio and flow velocity. In the third section, the analysis is related to the space ratio of 1.26, where the coherence between wake flow and cylinder vibration is investigated. The last section focuses on the case when both cylinders are free to vibrate, considering a fixed L/D = 1.26.

All the cases presented are compared to the results from the measurements with a single cylinder. The analysis with L/D = 1.26 was detailed due to the observed results comparing with higher space ratios.

*3.1. Tandem Configurations—Non-Dimensional Amplitude and Main Frequencies*

The vibration response with the cylinders is performed using non-dimensional amplitude and reduced velocity for cases with different values of L/D and the configurations first cylinder free to vibrate (FV), second cylinder free to vibrate (SV) and both cylinders free to vibrate (BV). Figure 3 presents the cases tested and low amplitudes are observed due to the high mass ratio and the selected natural frequencies. The reduced velocities are higher than 6 and lower than 75. The cylinder was free to vibrate transversally to the flow; the mass and damping ratios and the natural frequencies are described in Table 1.

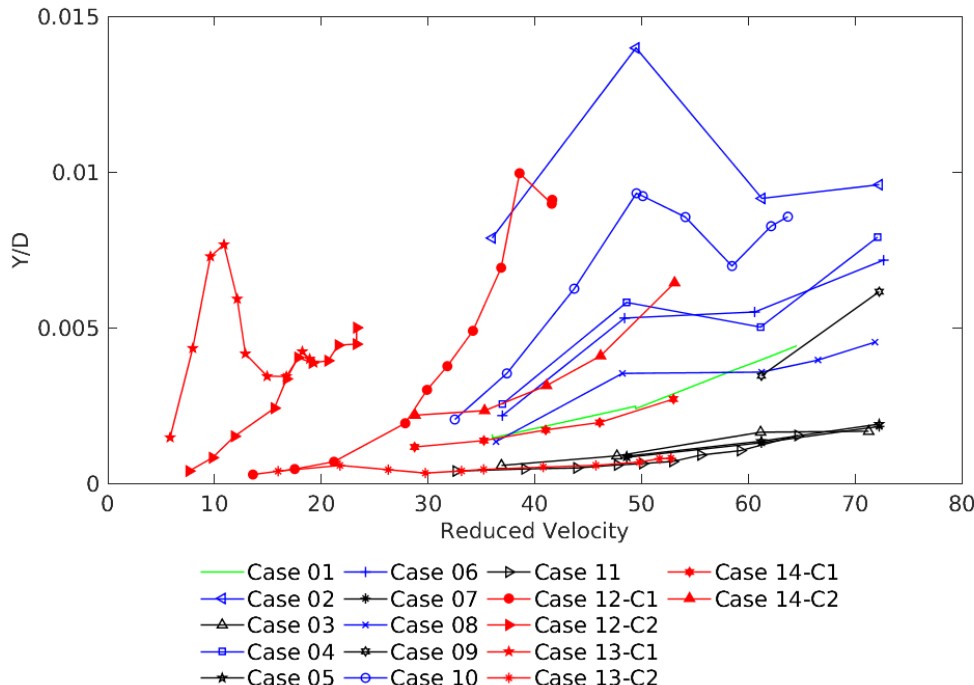

**Figure 3.** Tested cases: non-dimensional displacement (Y/D) versus reduced velocity (Vr).

The results for a single cylinder, Figure 3, show non-dimensional amplitude RMS increasing with the increase in the reduced velocity, which is a consequence of the main flow velocity increase. This response is equivalent to the results presented in [13]. Due to the high mass ratio and the high reduced velocities, the resulting amplitude values are low [20].

The analysis with the first cylinder free to vibrate (FV), Figure 3, presents higher values of non-dimensional displacement for L/D = 1.26 (Case 2), 1.4 (Case 4), and 1.6 (Case 6) than the single cylinder. In the case with L/D = 3.52 (Case 8), the first cylinder free to vibrate presents a similar non-dimensional displacement to the single cylinder. The higher non-dimensional amplitudes are observed in the L/D = 1.26 and decrease with the increase of the space between the cylinders. In the literature, the results presented by [9] showed that the non-dimensional displacement was low for a reduced velocity higher than 10. In [6] the high non-dimensional amplitudes for a cylinder with the same diameter and L/D higher than 5 was observed. From these analyses, it is expected that, in this configuration, the higher L/D will not influence the first cylinder wake formation, making the larger L/D to present an equivalent response with single cylinder.

The results with the second cylinder free to vibrate (SV), Figure 3, show lower amplitudes than the single cylinder at L/D = 1.26 (Case 3), 1.4 (Case 5), and 1.6 (Case 7). In the case of L/D = 3.52 (Case 9), the second cylinder free to vibrate presents similar non-dimensional amplitudes with the single-cylinder and increases with higher values of reduced velocity. In this configuration, the low amplitudes can be related to the proximity

of the cylinder and a non-interaction of the shear layers from the first cylinder with the second cylinder due to the close space ratio. The authors of [12] stated that as the separation distance between the cylinders increases, the amplitudes of the oscillation of the second cylinder increase over a wide range of reduced velocities. This response can be related to wake-induced vibration and observed in Figure 3.

The results from both cylinders free to vibrate are for L/D = 1.26 and indicate that the relation between the assembly characteristics of each cylinder is an important factor regarding the vibration response. In the results of Case 12, an increase in amplitudes on the second cylinder compared with the first cylinder can be observed. The results are equivalent in the initial reduced velocities and increase in C1 more than in C2. In this case, the first cylinder presents a lower natural frequency than the second cylinder. In the Case 13, the first cylinder presents higher amplitudes in the reduced velocities up to Vr = 12 that can be related to the vortex shedding process. After that the amplitude decrease. In Case 14, the response is closer but higher in the second cylinder; in this case both, cylinders present similar natural frequencies. The results for most cases remain between reduced velocities 20 and 72, but Case 12 (C2) and Case 13 (C1) start at a reduced velocity 6 due to the natural frequencies tested being higher than the additional cases tested.

In the literature, some studies observed similar results. The authors of [9] studied two cylinders with a space ratio of 1.2 and observed higher amplitudes on the downstream cylinder for high reduced velocities; this was observed with both cylinders free to vibrate and with just the second cylinder free to vibrate. In L/D = 1.5, the first cylinder vibrating showed higher amplitudes than the second cylinder vibrating but with lower values than that observed for small L/D. For the two cylinders free to vibrate in a tandem arrangement, [10] observed that the first cylinder free to vibrate increased the amplitudes for reduced velocities over 12 L/D = 1.57, while the second cylinder maintained low amplitudes.

Studies that investigated both cylinders free to vibrate include [7,9,10,21], or studies concerning the second cylinder free to vibrate are [22] and [11]. In all the cases, the results indicate that the increase in displacement occurs due to the influence of the first cylinder wake and due to the relation between the frequencies of the cylinders free to vibrate. The increase in displacement on the first cylinder with a close space, as observed in Figure 3, is not common in the literature. Even with small amplitudes, the phenomena are relevant to applications such as tube banks, where the change in the fixation of the tubes can represent an application similar to the present configuration, FV or SV, and the close space is a characteristic observed in tube banks. To better understand the influence of the space ratio and the position of a cylinder free to vibrate on the dominant frequencies on a cylinder free to vibrate, the frequency spectrum, continuous wavelet, cross-correlation, and wavelet coherence were applied in the present analysis.

### 3.2. Frequency Ranges and Flows Patterns in Tandem Configurations

The comparison between the frequency response in the time domain and frequency–time domain for the tandem arrangement L/D = 1.26 (Case 2 and Case 3); L/D = 1.4 (Case 4 and Case 5); L/D = 1.6 (Case 6 and Case 7); and L/D = 3.52 (Case 8 and Case 9) are presented in Figure 4. The results for the configuration with the first cylinder free to vibrate and the second cylinder free to vibrate are plotted. The results are presented for the reduced velocity of 72 for all the space ratios. The use of reduced velocities over 5, which is related to the vortex shedding, is due to the fluid elastic instability in tube banks that occur in reduced velocities over 30.

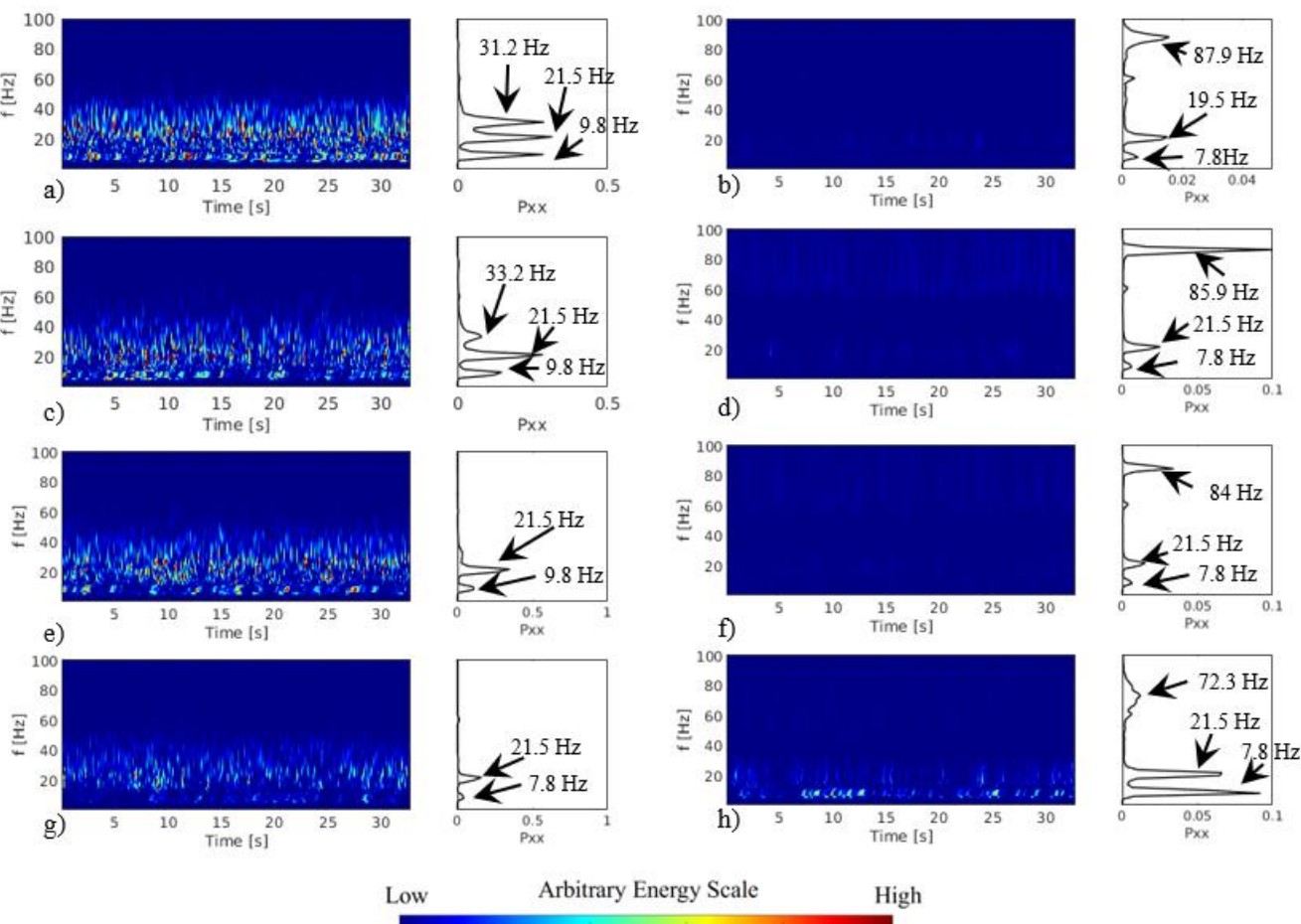

**Figure 4.** Continuous wavelet and Fourier spectrum from acceleration signals on the cases: (**a**) Case 2 with Vr = 72 for L/D =1.26, (**b**) Case 3 with Vr = 72 for L/D =1.26, (**c**) Case 4 with Vr = 72 for L/D =1.4, (**d**) Case 5 with Vr = 72 for L/D =1.4, (**e**) Case 6 with Vr = 72 for L/D =1.6, (**f**) Case 7 with Vr = 72 for L/D =1.6, (**g**) Case 8 with Vr = 72 for L/D = 3.52, (**h**) Case 9 with Vr = 72 for L/D = 3.52.

The results for L/D = 1.26 with the first cylinder free to vibrate are presented in Figure 4a (Case 2), wherein the power spectrum of three distinct peaks can be observed. The two first peaks are associated with the natural frequencies and the third with the excitation of the flow and could be associated with the vortex shedding. The results for L/D = 1.26 with the second cylinder free to vibrate (Case 3) are presented in Figure 4b. Two peaks linked to the natural frequencies are observed, the peak around 87.9 Hz can be linked to the excitations due to the increase in the flow velocity and the first cylinder wake. The response in the energy magnitudes and the distribution in the frequencies in Figure 4a shows high energy levels encompassing the three peaks observed in the power spectrum. In Figure 4b, the levels of energy decrease and spread in four main regions. The energy magnitudes are lower in the second cylinder than in the first cylinder. The higher non-dimensional amplitudes observed in Figure 3 can be related to the conditions of the test that concentrate the energy in the main frequency.

The results for L/D = 1.4 with the first cylinder free to vibrate are shown in Figure 4c, Case 4. The energy levels at the first and second peaks observed in the power spectrum can be associated with the natural frequencies. The third peak is observed at 33.2 Hz, which could be associated with the excitation of the flow. In the continuous wavelets, there is an increase in the energy spectrum associated with the first peak, mainly up to 15 s. The results for L/D = 1.4, with the second cylinder free to vibrate, are shown in Figure 4d, and three peaks of energy are observed in the power spectrum results: the first two peaks are associated with the cylinder's natural frequencies. The energy levels increased

mainly with the third frequency peak, which can be associated with turbulence excitations and an increase in the kinetic energy.

In the literature, some studies presented similar results. The authors of [9] studied two cylinders with a space ratio of 1.2 and observed higher amplitudes on the downstream cylinder for high reduced velocities; this was observed with both cylinders free to vibrate and with just the second cylinder free to vibrate. For L/D = 1.5, the first cylinder vibrating showed higher amplitudes than the second cylinder vibrating but with lower values than observed for small L/D. For the two cylinders free to vibrate in a tandem arrangement, [10] observed that the first cylinder free to vibrate increased the amplitudes for reduced velocities over 12 and L/D = 1.57, while the second cylinder maintained low amplitudes.

The results for the configuration of tandem cylinders with L/D = 1.6 with the first cylinder free to vibrate are presented in Figure 4e, Case 6. The main excitation frequencies remain associated with the natural frequencies range. The energy levels decrease and spread between the frequency ranges. The results for L/D = 1.6 with the second cylinder free to vibrate are presented in Figure 4f (Case 7). The first peak is related to the second natural frequency and the second peak could be related to the electric power network but, in additional tests, this frequency increases with the flow velocity increase, since it is associated with the flow excitation

The results for the tandem configuration L/D = 3.52 with the first cylinder free to vibrate are presented in Figure 4g (Case 8). Two peaks associated with the natural frequencies are observed. For L/D = 3.52, with the second cylinder free to vibrate (results are in Figure 4h, Case 9), the energy increases and the third peak changes to 72.3 Hz and is associated with the increase in the velocity flow.

The results presented from L/D = 1.4, Figure 4c,d, show similar responses to the results for L/D = 1.26, Figure 4a,b. The main frequencies are in the same range, but the energy level is lower with the increase in the space ratio. For the first cylinder free to vibrate, it can be observed that the peak of frequency associated with the first natural frequency is higher and could be related to the mass-damping parameters and the pressure field that generates an excitation frequency close to the natural frequency.

The authors of [2] presented an analysis of the mean pressure distribution around tandem fixed cylinders for several space ratios L/D. For the space ratio of L/D = 1.2, the author observed higher pressure coefficients on the first cylinder than in the second one. It was also observed that on the second cylinder, the pressure coefficients have asymmetric distribution along time due to the wake of the first cylinder. The increase in vibration of the first cylinder observed in this work could be related to the pressure distribution due to the second cylinder fixed and inside the wake.

In the results for L/D = 1.6, Figure 4e, the energy level decrease compared with the smaller space ratios and the energy concentrates on the two natural frequencies. In Figure 4f, the energy levels are lower than the ones observed for small space ratios, but the ranges of frequency remain the same. Comparing the results from L/D = 3.52 with the lower space ratios, it can be observed that for the first cylinder free to vibrate, energy levels must decrease significantly (Figure 4g). For the second cylinder free to vibrate and L/D = 3.52, an increase in energy associated with the natural frequencies can be observed.

The authors of [8] studied the wake-induced vibration in an L/D = 4 tandem cylinder set. In their numerical results for low Reynolds numbers, they observed a large amplitude of the downstream cylinder associated with a low-frequency component of the transversal load, which was closer to its natural frequency. The authors of [7] tested both cylinders free to vibrate and observed that the second cylinder underwent a resonance-like response as the flow velocity increased and did not stabilize until the highest reduced velocity for small and moderate spacings (L/D = 3.5 to 10). The second cylinder's oscillation frequency in the transverse direction was not affected by the spacing variations, which was a result of the interaction between fluid and the cylinder itself. The authors of [9] tested cases with L/D = 3 and L/D = 4, with lower amplitudes than for the lower L/D tested in this paper.

For L/D = 3, [2] observed that for tandem-fixed cylinders, higher pressure coefficients in the second cylinder and an asymmetric distribution compared with lower space ratios; due to the distance from the first cylinder, its wake impinged on the second cylinder. The authors of [11] showed that by increasing the mass-damping parameter, the amplitude decreases and the range of lock-in becomes narrow. For small space ratios, lower amplitudes were observed.

The flow patterns observed in visualizations and the monitoring of pressure or force in the literature show similarities; a scheme of the flow over tandem cylinders with variation in the space ratio tested is shown in Figure 5. The scheme derives from the results from fixed cylinders and the pressure distribution presented in the literature [1,2,23,24]. The discussion is linked to its impact on the vibration response.

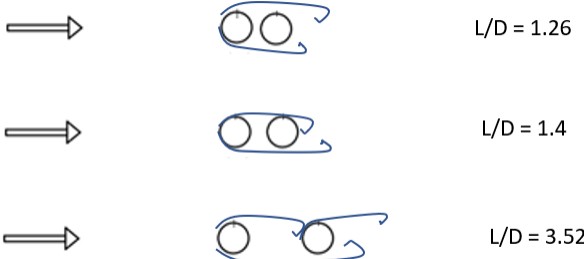

**Figure 5.** Schematic flow patterns over tandem arrangements with L/D 1.26; L/D 1.4; and L/D 3.52.

For L/D = 1.26 (Figure 5), it is observed that the wake from the first cylinder encompasses the second cylinder due to the close space, this characteristic generates a mean pressure field symmetric [2] in the second cylinder and can be related to the observed increase in displacement observed in Figure 3. The frequency peaks around the natural frequencies of the first cylinder can be associated with the small distance between the cylinders and non-interaction of the shear layers from the first cylinder with the second cylinder. On the second cylinder free to vibrate, the vortex shedding frequency is observed due to the changes in lift force during the vortex formation.

For the L/D = 1.4 arrangement, the flow pattern can allow the reattachment of the shear layer on the second cylinder creating an asymmetric pressure field. The distance between cylinders in the L/D = 3.52 configuration allows the formation of a vortex street between the cylinders that impinges on the second cylinder in the tandem configuration, generating an excitation frequency linked to the wake from the first cylinder.

The visualization presented in [23] for L/D = 1.26 and [24] for L/D = 2.5 and L/D = 5 showed the different flow pattern around tandem cylinders.

The study of the frequency ranges of excitation for all the conditions tested is presented in Figure 6, where the results from the power spectrum analysis were compiled and plotted. The analysis is presented using the acceleration signals obtained in the cylinder free to vibrate. The natural frequencies are represented with vertical lines for 7.8 Hz and 21.5 Hz. The tested cases are presented for each reduced velocity with the energy level obtained from the power spectrum results. In the x-axis, the frequency and the corresponding Strouhal number are presented. The Strouhal number is determined with the reference velocity and the diameter of one cylinder.

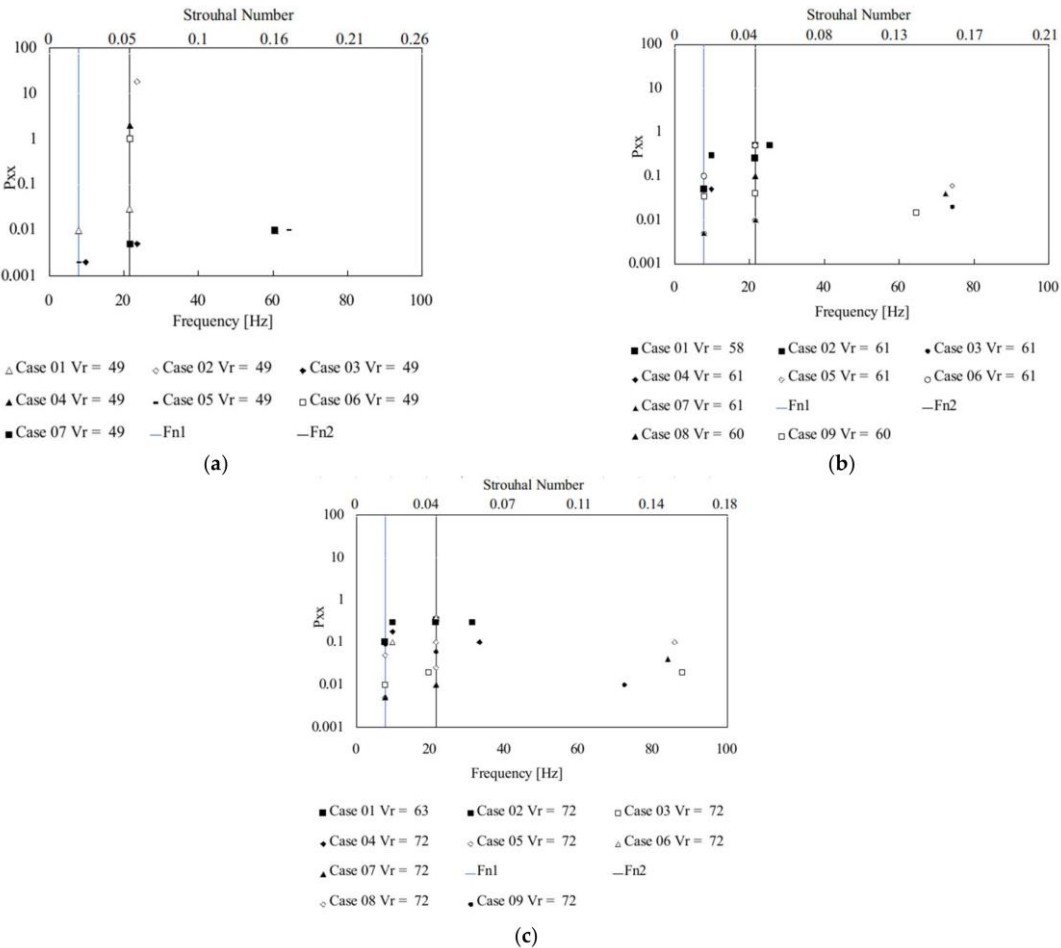

**Figure 6.** Frequency peaks observed in the Fourier spectrum from acceleration signals for the tested cases with FV and SV in (**a**) reduced velocity around Vr = 49, (**b**) reduced velocity around Vr = 60, and (**c**) reduced velocity around Vr = 70.

In a general analysis, it can be observed that all the cases concentrate peaks around the natural frequencies, but results between 60 Hz and 90 Hz are also visible, depending on the reduced velocity. In Figure 6a, the results obtained in reduced velocity, Vr, equal to 49, show the highest levels of energy linked to the second natural frequency for L/D = 1.26 and with the second cylinder free to vibrate. The second cylinder free to vibrate shows frequency peaks around 60 Hz to L/D = 1.26, 1.4, and 1.6 and can be associated with St = 0.16 and St = 0.17. In Figure 6b, in the results for reduced velocity, Vr, around 60, the frequencies remain to concentrate around the natural frequencies. The second cylinder free to vibrate presents frequency peaks between 60 Hz and 80 Hz, generating a Strouhal number between 0.13 and 0.17. In Figure 6c, the results with reduced velocity, Vr, around 70, show that the second cylinder free to vibrate remains with a peak of frequency between 60 Hz and 100 Hz, which is related to the increase in the flow velocity. The Strouhal numbers observed remain between St = 0.13 and St = 0.16.

The Strouhal numbers obtained from the first cylinder free to vibrate are lower than expected for the tandem configuration tested in [2,3]. The frequency peak observed in the power spectrum can be an excitation not directly associated with the vortex shedding. Due to the high interaction between the cylinders at a close space ratio, the flow excitation was able to maintain the response around the natural frequency once a non-clear excitation occurred in the first cylinder due to the presence of the second cylinder inside the wake. The flow excitation can also be influenced by the mass-damping parameter of the free to vibrate cylinder, altering the FIV-response.

The Strouhal numbers obtained from the second cylinder free to vibrate agree with the literature [2,3], being associated with the wake from the first cylinder. Even for the closest space tested cases, it can be observed the shear layer interact between the cylinders due to the vortex formation.

### 3.3. Tandem Configuration L/D = 1.26—Coherence between the Wake Velocity and the Acceleration

The results in Figures 3 and 6 showed that the close space impacts the first cylinder vibration response, which is a different mechanism than the wake-induced vibration observed in higher space ratios, where the second cylinder present higher amplitudes. To understand this behavior, a study with hot wire and an accelerometer were utilized for L/D = 1.26, changing the cylinder free to vibrate position, the first cylinder free to vibrate (Case 10), and the second cylinder free to vibrate (Case 11). The Fourier spectrum of the wake velocity signal for both cases remain an equivalent frequency range, between 50 Hz and 100 Hz, as observed in Figure 7a,b; but in Case 11, the adjacent regions present lower energy. In both cases, the hot wire anemometer was positioned after the second cylinder, as presented in Figure 2c.

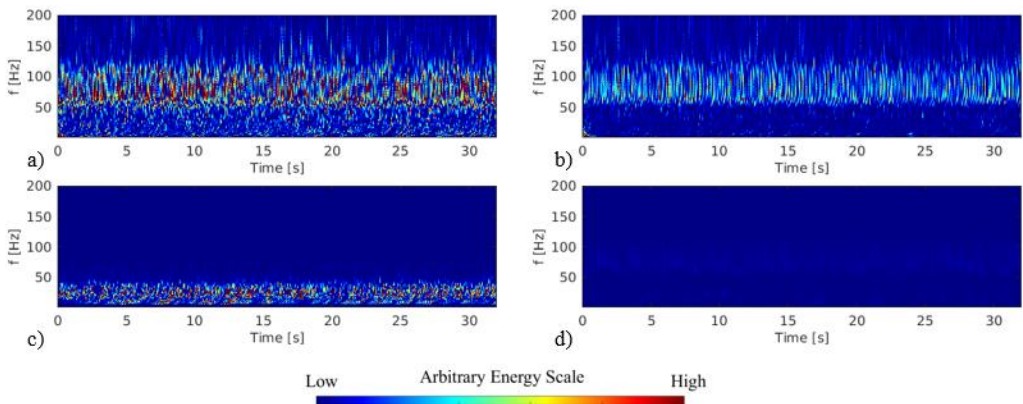

**Figure 7.** Continuous wavelet transforms of (**a**) Case 10—Signal of the wake velocity behind the first cylinder free to vibrate, (**b**) Case 11—Signal of the wake velocity behind the second cylinder free to vibrate, (**c**) Case 10—Acceleration signal of the first cylinder free to vibrate and (**d**) Case 11—Acceleration signal of the second cylinder free to vibrate.

The acceleration spectra, in Figure 7c,d, show a change in the frequency ranges where the energy level is higher. In the result from the first cylinder free to vibrate, Figure 7c, the energy concentrates in the range under 50 Hz that encompasses the cylinder natural frequencies. In Figure 7d, the levels of energy are low, but the region of excitation is between 50 Hz and 100 Hz, the same as observed in the wake velocity signals.

To investigate this relation, the coherence analyses were applied to the signals of wake velocity and acceleration for the Case 10 and Case 11 with the reduced velocity 63. The results are presented in Figure 8a for the first cylinder free to vibrate (Case 10) and 8b for the second cylinder free to vibrate (Case 11). The coherence analysis applies the coherence function with values between zero, which represents no coherence to a value of 1, which represents a complete coherence between the signals. As the coherence values decrease, the results can be associated with non-linear systems [16]. This analysis allows signals to be related and to identify whether they present a frequency range of direct relation.

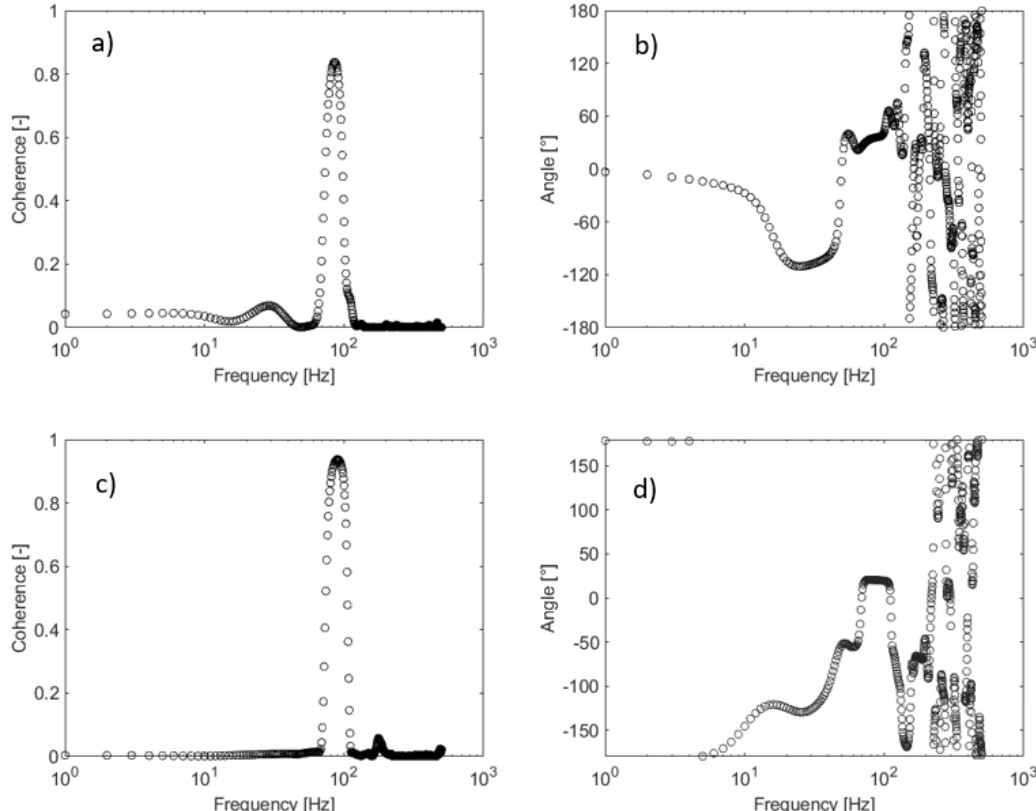

**Figure 8.** (**a**) First cylinder free to vibrate (Case 10)—Cross-coherence between wake velocity and cylinder acceleration for the reduced velocity equal to 63; (**b**) Phase angle of Case 10. (**c**) Second cylinder free to vibrate (Case 11)—Cross-coherence between wake velocity and cylinder acceleration for the reduced velocity equal to 63 and (**d**) phase angle of Case 11.

In Figure 8a, a region of high coherence, around 90%, between 80 Hz and 90 Hz is observed and a peak with around 10% coherence is observed around 30 Hz. A level in the phase angle of 120° can be observed between 20 Hz and 50 Hz in Figure 7b, demonstrating that there is a relationship between the signals in the second natural frequency range. Between 70 Hz and 100 Hz, a region around 40° shows a stabilization tending to a level but not in a single-phase angle. The random phase angle response, over 100 Hz, can be linked to higher frequencies in the turbulence spectrum.

In Figure 8c, the region of high coherence is between 90 Hz and 100 Hz and a second region with about 10% coherence is observed between 100 Hz and 110 Hz. The phase angle, in Figure 7d, presents a level of around 10° between 60 Hz and 100 Hz and a random phase angle response over 110 Hz. The high coherence is linked to the flow excitation, and a Strouhal number of around 0.17. This response represents the relation between the signals but not the strength of this relation (levels of energy).

The wavelet coherence is applied due to the non-stationary characteristic of the signals. The wavelet coherence analysis can provide insights into how two responses have evolved in time and frequency. The results of wavelet coherence show the color pallet representing red for one (100% coherence) and blue for zero (no coherence). The arrows represent the phase angle between signals for coherence values higher than 0.75. The wavelet coherence is shown in Figure 9a, representing the result of the wake velocity signal and the acceleration. A region of high coherence is observed between 90 Hz and 120 Hz. The regions between 8 and 16 Hz occasionally show high coherence (8 s, 12 s, and 28 s). The region between 16 Hz and 32 Hz shows high coherence in some positions.

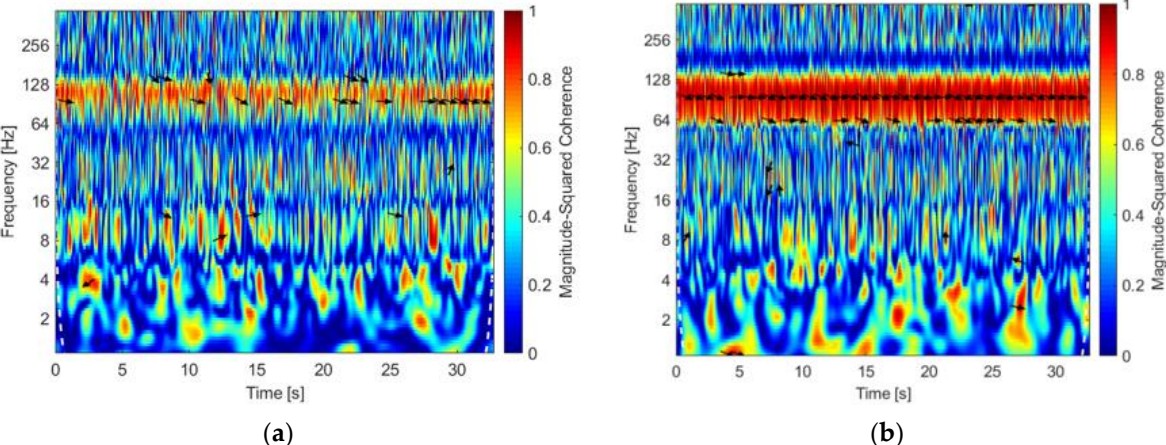

**Figure 9.** Wavelet coherence between wake velocity and cylinder acceleration for the reduced velocity equal to 63. (**a**) First cylinder free to vibrate and (**b**) second cylinder free to vibrate.

In Figure 9b, the coherence remains over 75% between 64 Hz and 128 Hz for all the time analyzed. The phase regions with lower frequencies show punctual regions as between 4 and 16 Hz around 15 s and between 16 and 32 Hz before 10 s, this region can be linked to the natural frequencies. The response observed in Figures 8 and 9 shows that the shedding frequency is presented in both situations tested, but it is not the main contribution to the increase in amplitude for the L/D 1.26, because the high energy in those cases is related to the natural frequency, as observed in Figures 4 and 7. The region of natural frequency shows a low coherence in the case with the first cylinder free to vibrate and can be a consequence of the movement observed in the first cylinder that is free to vibrate. In the second cylinder free to vibrate, the coherence between the vortex shedding and acceleration is close to the maximum due to the direct interactions of the cylinder movement and the wake; this response was expected due to the energy response observed in Figure 4, where a region of excitation associated with the vortex shedding was also observed.

The response observed can be related to results presented by [1], where the pressure distribution around cylinders in tandem with L/D = 1.2 was presented and in the first cylinder showed low-pressure coefficients after 90°; this is also observed between 0° and 50° in the second cylinder. The pressure fluctuations are high in the second cylinder between 60° and 120°. The region of low-pressure coefficients occurs between the cylinder and influences the fluctuations of the lift force. The fluctuations of pressure and force are low, creating a small movement in the cylinder around the natural frequencies. This force is not high enough to result in a significant amplitude but causes the cylinder to move at a frequency near to the natural, due to the close space.

### 3.4. Tandem Configuration L/D = 1.26—Two Cylinders Free to Vibrate

### 3.4.1. Two Cylinders Free to Vibrate with the Same Natural Frequency Range

The analysis of the Case 14, where both cylinders were free to vibrate was investigated using close natural frequencies for each cylinder in the assembly and applying the continuous wavelet transform (CWT), cross-coherence, and wavelet coherence. The CWT results are presented in Figure 10, showing an increase in the energy levels with the flow velocity. The energy levels are higher in the second cylinder free to vibrate for all the conditions showing a response different than the one observed when only one of the cylinders is free to vibrate, see Figure 4a,b, where the first cylinder free to vibrate presents higher displacement. This response indicates that when both cylinders are free to vibrate, with natural frequencies in the same range, this movement results in an energy reduction of the first cylinder but increase the interaction between the shear layers from the first with the second cylinder.

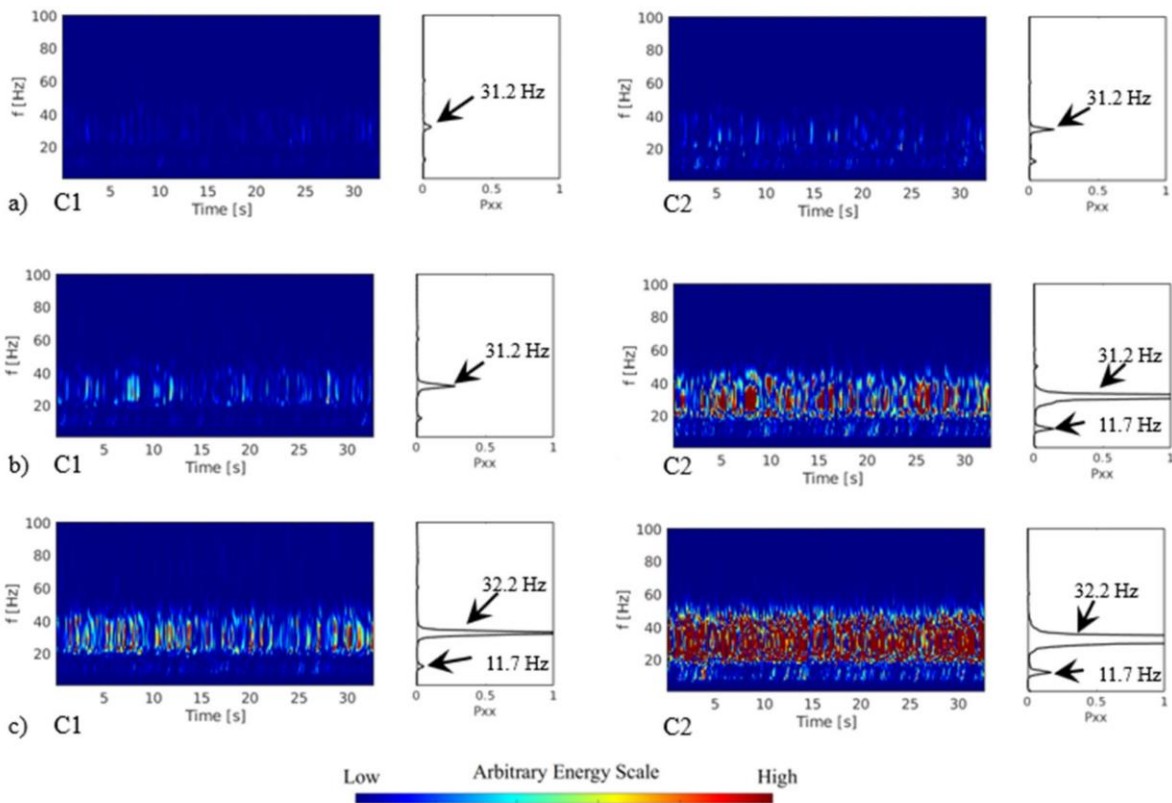

**Figure 10.** Continuous wavelet and Fourier spectrum of acceleration signals of the Case 14 with two cylinders free to vibrate: (**a**) reduced velocity = 35, (**b**) reduced velocity = 46, and (**c**) reduced velocity = 53.

The cross-coherence between the signals of acceleration from each cylinder is shown in Figure 11a and the phase angle in Figure 11b. The acceleration signals were measured simultaneously. In Figure 11a, one region with about 40% coherence is observed between 20 and 40 Hz and regions with small coherence are observed up to 300 Hz. The phase angle, in Figure 11b, shows a stable level around 150° from 20 Hz to 40 Hz, showing a clear relationship between the signals. This range of frequencies encompass the second natural frequency peak obtained from the assembly analysis and is the region with higher energy in all the results in Figure 10.

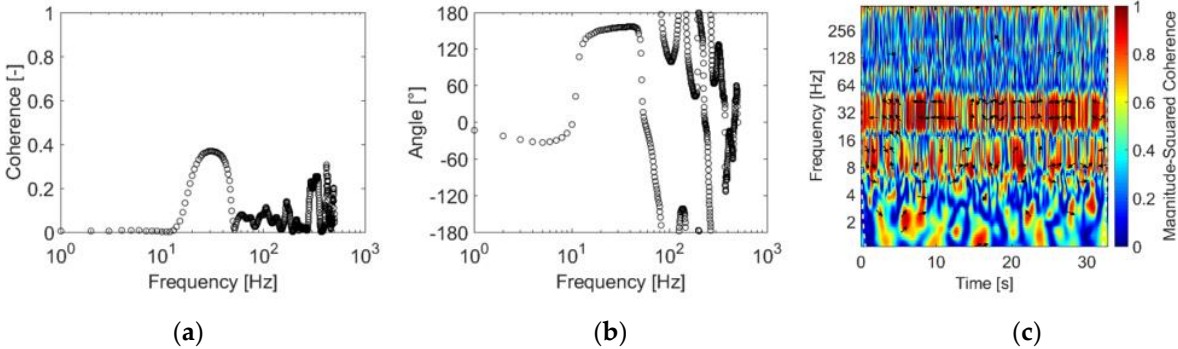

| (**a**) | (**b**) | (**c**) |
|---------|---------|---------|

**Figure 11.** (**a**) Cross-coherence between acceleration signals from both cylinders free to vibrate for reduced velocity equal to 46, (**b**) phase angle obtained in the cross-coherence, and (**c**) wavelet coherence between acceleration from the first cylinder and the second cylinders free to vibrate for reduced velocity equal to 46.

In Figure 11c, the wavelet coherence between acceleration from the first cylinder free to vibrate and the second cylinder free to vibrate for a reduced velocity equal to 46 is

presented. Two main regions of high coherence are observed, one between 8 Hz and 16 Hz associated with the first natural frequency and the second one between 20 Hz and 50 Hz associated with the second natural frequency. In Figure 11, it is observed that the coherence is higher in some instants of times as around 5 s, 8 s, the 20 s, and 25 s between 20 Hz and 50 Hz.

The results showed that the cylinders present excitation related to the natural frequencies, due to the proximity in the assembly because of the flow excitation. The authors of [8] observed a large displacement of the downstream cylinder for cases with L/D > 4 and linked it to the appearance of a low frequency component in the transverse load, which is closer to the natural frequency of the downstream cylinder.

### 3.4.2. Two Cylinders Free to Vibrate—The First Cylinder with Higher Natural Frequency than the Second

In the analysis of Case 13, the natural frequency of cylinder C1 is three times higher than the second cylinder. The CWT results of cylinder acceleration for more than a reduced velocity are shown in Figure 12. The estimated reduced velocity over each cylinder is presented, the values for the first cylinder are lower due to the natural frequency value higher than the second cylinder. Figure 12 presents the results for different flow velocities (a, b, and c) and cylinders (C1 and C2).

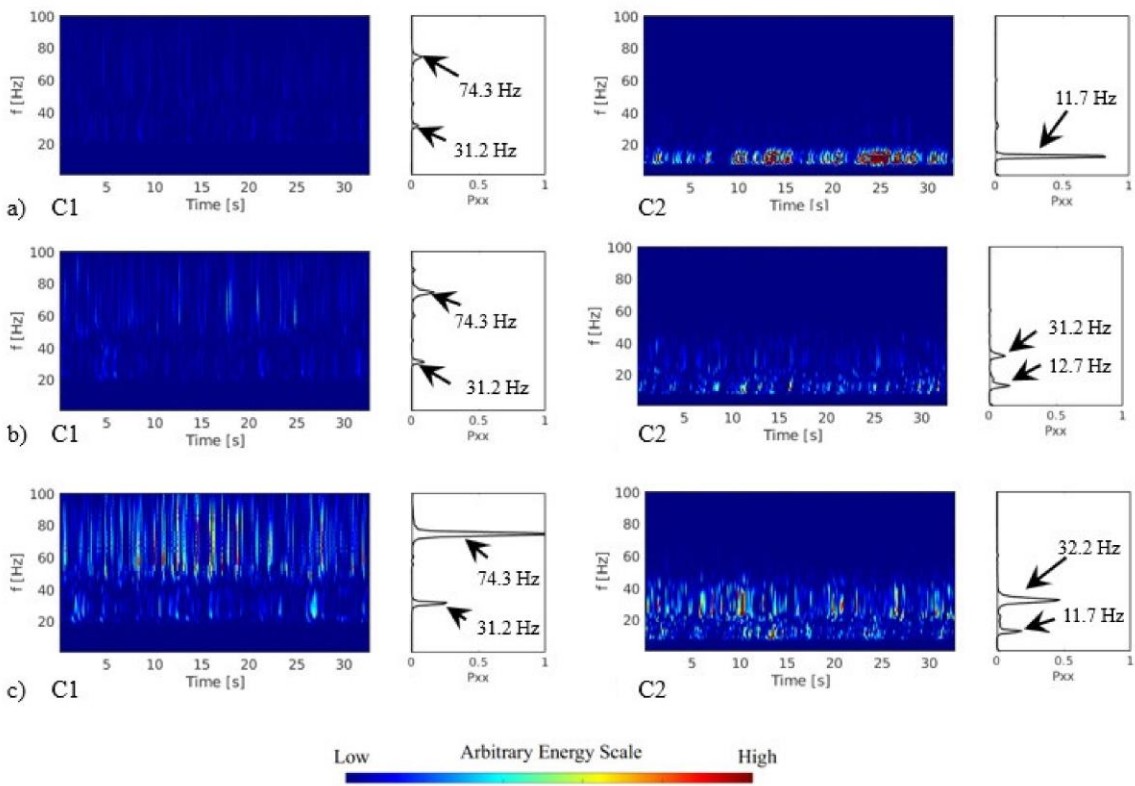

**Figure 12.** Continuous wavelet and Fourier spectra of acceleration signals of the case with two cylinders free to vibrate: (**a**) C1—first cylinder free to vibrate, Vr = 11; C2—second cylinder free to vibrate, Vr = 30; (**b**) C1—first cylinder free to vibrate, Vr = 15; C2—second cylinder free to vibrate, Vr = 41; (**c**) C1—first cylinder free to vibrate, Vr = 18; and C2—second cylinder free to vibrate, Vr = 52.

In Figure 12a, it can be observed that the energy level of the acceleration of the second cylinder is higher close to the natural frequency. The frequency ranges with higher energy change from the first to the second cylinder, with influence from the flow observed in the results of the first cylinder.

In Figure 12b, the levels of energy remain unaltered. In the results from the first cylinder, a range between 20 Hz and 40 Hz is associated with the first natural frequency, and the second peak is around 80 Hz and can be associated with the vortex shedding. The second cylinder presents two peaks related to the natural frequencies of the cylinder.

In Figure 12c, the first cylinder presents an increase in energy on the frequency range linked to the vortex shedding; in the second cylinder the levels of energy increase around the natural frequencies. The amplitudes in Figure 3 show this increase of displacement associated with the energy in C1.

A cross-coherence analysis performed using the acceleration signals obtained simultaneously from both cylinders free to vibrate is shown in Figure 13 with the corresponding phase angle.

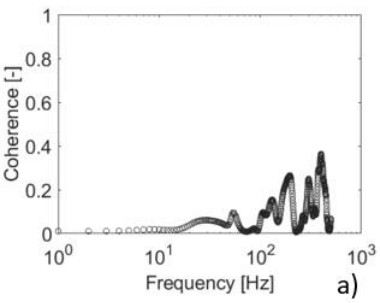 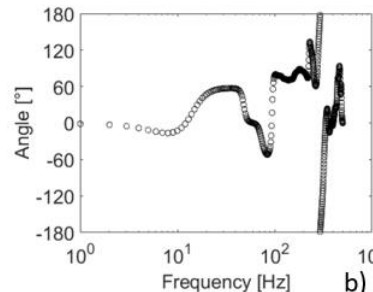 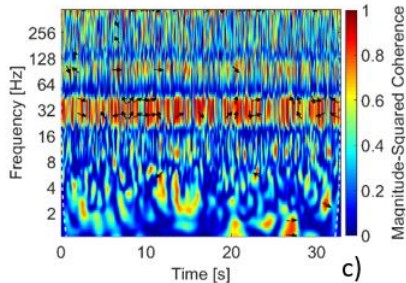

**Figure 13.** (**a**) Cross-coherence of the acceleration signals from both cylinders free to vibrate for reduced velocity equal to Vr = 18 for the first cylinder and Vr = 52 for the second cylinder, (**b**) phase angle obtained in the cross-coherence, and (**c**) wavelet coherence of the acceleration from the first cylinder and acceleration from the second cylinders free to vibrate for equal reduced velocity.

The results in Figure 13 show, in general, no coherence between the signals. The range between 20 Hz and 40 Hz has coherence values lower than 10%, with a phase angle of around 60°. Above 100 Hz, the coherence values are around 25%, with a phase angle of 75°.

In the wavelet coherence, seen in Figure 13c, the region with higher coherence is between 20 Hz and 40 Hz, which was observed in all the results presented in Figure 12. A region with intermittent high values of energy is observed in the range between 60 Hz and 110 Hz.

### 3.4.3. Two Cylinders Free to Vibrate—The First Cylinder with Lower Natural Frequency than the Second

This section presents the analysis where both cylinders are free to vibrate; the first cylinder presents a natural frequency two times lower than the second cylinder (Case 12). The results with CWT and with the increase in the flow velocity are presented in Figure 14. The estimated reduced velocity over each cylinder is presented, the values for the first cylinder are higher due to the natural frequency value being lower than the second cylinder. The results in Figure 14 are divided by the flow velocity (a, b, and c) and by the cylinder (C1 and C2).

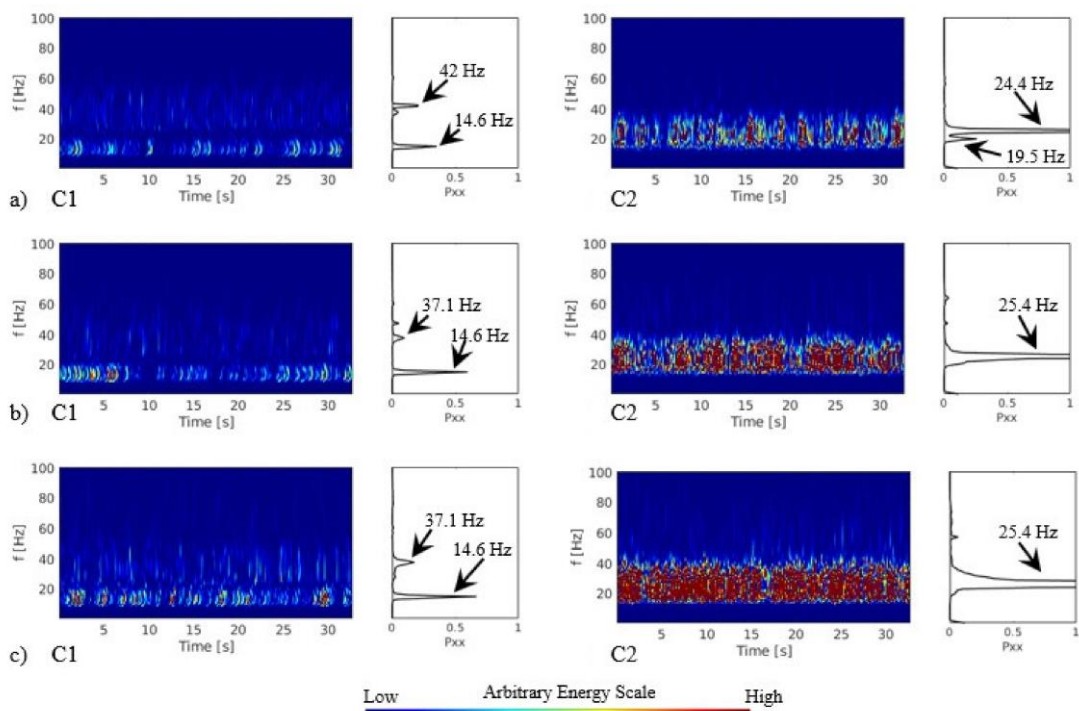

**Figure 14.** Continuous wavelet and Fourier spectra from the acceleration signals on the case with two cylinders free to vibrate (**a**) Vr C1 = 30, Vr C2 = 17; (**b**) Vr C1 = 34, Vr C2 = 19; (**c**) Vr C1 = 41, Vr C2 = 23.

In Figure 14a, the levels of energy, linked to the natural frequencies, have higher values for the second cylinder. The increase in the flow velocity, Figure 14a–c, shows that the second cylinder presents an increase in the energy levels and that is why the amplitudes are higher in this configuration.

The analysis of cross-coherence was performed with the acceleration signals from each cylinder. The coherence result and the phase angle are presented in Figure 15a,b. The coherence is low in the range of up to 120 Hz.

Figure 15c shows the wavelet coherence of the acceleration of the first and the second cylinder free to vibrate. The main region of high coherence is between 8 Hz and 16 Hz associated with the natural frequency of the first cylinder. The coherence is higher in the frequency range that is equivalent in both cylinders.

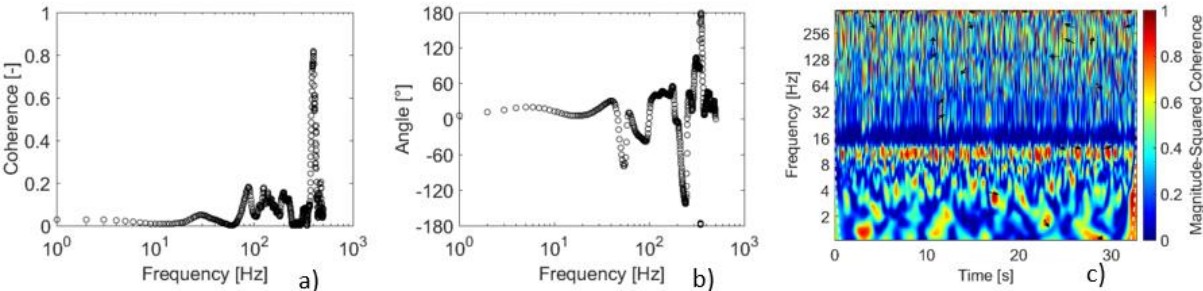

**Figure 15.** (**a**) Cross-coherence of the acceleration signals from both cylinders free to vibrate for reduced velocities equal to Vr C1 = 41 and Vr C2 = 23, (**b**) phase angle obtained in the cross-coherence, and (**c**) wavelet coherence of acceleration from the first cylinder and acceleration from the second cylinders free to vibrate for reduced velocities equal to Vr C1 = 41 and Vr C2 = 23.

The higher amplitudes in the cylinder with lower natural frequency can be associated to the energy necessary to move the cylinder with higher natural frequency, which in this case can be related to the stiffness of the assembly.

## 4. Conclusions

An experimental investigation of the influence of the space ratio of two cylinders in tandem arrangement on the excitation frequencies of one or both cylinders free to vibrate was performed. The conditions studied were the first cylinder free to vibrate, the second cylinder free to vibrate, and both cylinders free to vibrate. The space ratio values tested were 1.26, 1.4, 1.6, and 3.52.

The investigation using low space ratios is important to understand the main mechanisms of excitation observed. This information allows the study of the main contributions of each variable in more complex arrangements such as tube banks. The tested conditions aim to investigate the close space influence, while most of the literature presents larger space ratios due to the interest in wake-induced vibration.

The study with changes in the space ratio was analyzed for the configurations of the first cylinder free to vibrate and the second cylinder free to vibrate. The first cylinder free to vibrate showed that higher displacement is observed for the space ratio of 1.26, and it decreases with the increment in the space ratio. The main frequencies observed in the cases with the first cylinder free to vibrate remain near the natural frequencies.

In the analysis of the second cylinder free to vibrate, the higher displacement is observed for the space ratio of 3.52, which can be associated with the wake-induced vibration. The frequency analysis for the second cylinder showed that a frequency around the vortex shedding mechanism is observed in the majority of the cases, but the energy in this frequency range increases in a significant way only in the highest L/D tested.

The flow patterns observed in the literature for the tested L/D allow the association of the observed response to the interaction between the shear layer of the first cylinder and the second cylinder. The movement pattern observed is not linked to any frequency region related to flow excitation. The high energy levels were in the natural frequency ranges, and this response could be related to the changes in the flow pattern and as a consequence of the cylinder movement without a clear frequency and the system response on the natural frequency range.

The increase in the displacement in the close space arrangement (L/D = 1.26) when the first cylinder is free to vibrate was investigated by analyzing the wake response and the corresponding cylinder acceleration. The cross-coherence results showed that for the first and the second cylinder free to vibrate, a high coherence was observed for the range of 80 Hz to 100 Hz associated to the vortex shedding, even though energy levels analyzed with wavelet transforms do not show significant concentration in this range. The wavelet coherence also showed higher relation in the range of 64 Hz to 128 Hz.

For L/D = 1.26, the vibration pattern was investigated in cases where both cylinders were free to vibrate considering that they have equal or different natural frequencies. In the case where both cylinders have equivalent natural frequency, it was observed that the response changed in the cases where just one cylinder is free to vibrate. The results showed that the energy level of the signal from the second cylinder increased more than that from the first cylinder. For the case where the first cylinder's natural frequency is higher than the second, a response similar to the cases where the second cylinder was fixed was observed. In the case where the first cylinder's natural frequency is lower than that of the second cylinder, the increase in energy of the second cylinder occurs, close to the response observed in wake-induced vibrations. The results presented for both cylinders free to vibrate shows the dependency of assembly characteristics and this can affect which cylinder will become unstable in an assembly of the cylinders.

In all cases of tandem cylinders tested, with one or both cylinders free to vibrate, the main frequency ranges were related to the natural frequencies of the cylinders. Additional excitation frequencies were observed, but they do not influence a response, such as excitations around natural frequencies, which may be related to pressure and velocity fluctuations due to disturbed flow. The close space alters the flow structures and the shear layer interaction that excites the cylinder.

**Author Contributions:** Conceptualization, R.F.N., A.P.P. and S.V.M.; methodology, S.V.M., R.F.N.; software, R.F.N., A.P.P.; validation, R.F.N., A.P.P. and S.V.M.; formal analysis, R.F.N.; investigation, R.F.N.; resources, S.V.M.; data curation, R.F.N.; writing—original draft preparation, R.F.N.; writing—review and editing, S.V.M. and A.P.P.; visualization, R.F.N.; supervision, S.V.M.; project administration, S.V.M., funding acquisition, S.V.M. All authors have read and agreed to the published version of the manuscript.

**Funding:** This research was funded by CNPq, Conselho Nacional de Desenvolvimento Científico e Tecnológico, grant number 312133/2021-9"

**Data Availability Statement:** The data used in the present analysis are available in: Neumeister, R. F.; Petry, A. P.; Möller, S. V. 2022-a. "Experimental data of the space ratio influence on the excitation frequencies of one and two cylinder free to vibrate in tandem arrangement", Mendeley Data, V2, doi: 10.17632/33rf3kffb2.2.

**Conflicts of Interest:** The authors declare no conflict of interest. The funders had no role in the design of the study; in the collection, analyses, or interpretation of data; in the writing of the manuscript; or in the decision to publish the results.

**Nomenclature**

| Symbol | Definition |
| --- | --- |
| C1 | First Cylinder |
| C2 | Second Cylinder |
| D | Diameter [m] |
| Db20 | Daubechies Wavelet |
| f | Frequency [Hz] |
| $f_{n1}$ | First Natural Frequency [Hz] |
| $f_{n2}$ | Second Natural Frequency [Hz] |
| g | Gravity [m/s²] |
| L | Longitudinal Pitch [m] |
| L/D | Longitudinal Space Ratio [-] |
| m | Mass per Length [kg/m] |
| $m^* = \dfrac{4m}{\rho\pi D^2}$ | Mass Ratio [-] |
| Pxx | Power Spectrum |
| Re=UD/v | Reynolds Number [-] |
| St=fD/U | Strouhal Number [-] |
| t | Time [s] |
| U | Reference Velocity [m/s] |
| Vr=U/(D$f_{n1}$) | Reduced Velocity [-] |
| Y | Displacement Amplitude [m] |
| Y/D | Non-Dimensional Displacement [-] |
| Greek Letters | |
| $\zeta$ | Damping ratio [-] |
| v | Kinematic Viscosity [m²/s] |
| $\varrho$ | Density [kg/m³] |
| $\pi$ | Pi Number |

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
