# Peer review of "Experimental Analysis of the Space Ratio Influence on the Excitation Frequencies of One and Two Cylinders Free to Vibrate in Tandem Arrangement"

_vibration, doi:10.3390/vibration5040045_

Round 1

Reviewer 1 Report

This paper presents an extensive experimental study of an interesting problem in engineering. The results are analysed in an adequate manner and compared with the relevant literature. The work could be much more interesting and complete if a mathematical model (or other type or transferrable model) was derived from the experimenal findings in order to be able to extroplate the results into other setups. Also some practical applications of the experimental findings could be added. That being said, these two studies maybe could be developed in a 'sister' paper.

Author Response

Thank you for the comments and suggestions. We hope to work on a model including data from this paper in a near future.

Reviewer 2 Report

The paper presents experimental results concerning oscillations of two cylinders in tandem arrangement. The authors carefully describe the experimental procedure and results, compare them with other works. Before the publication I would like to suggest some major and minor revisions.

Major:

1.       The first eigenfrequency in Table 1 and in Figure 4 do not coincide. What is the reason?

2.       The plots for cases 05 and 12-C2 in Figure 3 are very different in comparison with the others. How can you explain it? Probably, it requires deeper analysis, if the reasons are physical.

3.       Two peaks in spectra in Figure 4 are associated with the natural frequencies of the cylinders. The third one is explained by different reasons (“the vortex shedding”, “excitations due to the increase in the flow velocity”, “turbulence excitations”,…). Are these reasons really different? Or they are different descriptions of the same phenomenon? If the reasons are different, please, let us know how you detected them in different cases.

4.       The peak in Figure 4d at 85.9 Hz has very high value of the q-factor. Could it be a sign of an autogeneration process? Please, give your opinion about this result.

5.       Could you underline (for example, in the conclusion) the result, which are different from the previously published, and evaluate their importance for understanding of the studied problem?

6.       It is necessary to give a brief analysis of the influence of obtained results on investigation in the field.

7.       I think that the abstract may contain less details, but more conceptual points including reflections about two last remarks.

Minor:

1.       To make clearer the experiment description I recommend to add

·         the scheme of the cylinder’s attachment because Figure 2b is not clear;

·         the sketch view of the cylinder’s oscillations corresponding to the first and second natural frequency;

·         direction of vibration measured by the accelerometer.

2.       It would be useful to add clear definition of the mass ratio and the damping ratio. How did your adjusted them in the experiment?

3.       Colored figures 4, 7, 10, 12 and 14 would be better for understanding.  

4.       The legend in Figure 3 seems to be chaotic. Please, rearrange names of curves. Also colored curves would be clearer.

Author Response

Thank you for your suggestions. We hope to clarify your questions.

Major:

  1. The first eigenfrequency in Table 1 and in Figure 4 do not coincide. What is the reason?

The main reason for this difference is that the natural frequency presented in Table 1 is measured in still air, and the results in Figure 4 and additional Figures are obtained from signals of vibration as a consequence of the flow. The flow influences the damping parameter and the added mass (Blevins, 1990), but this influence is difficult to identify and will change with the increase in the flow velocity and with the structures of the flow over the bluff body, that is why the measurements in still air are considered as reference.

Blevins, R. D. 1990. Flow-induced vibration. New York.

  1. The plots for cases 05 and 12-C2 in Figure 3 are very different in comparison with the others. How can you explain it? Probably, it requires deeper analysis, if the reasons are physical.

Thank you for your comment. Due to the comment about the differences between the two curves in relation to the additional cases and the difficult reading of the figure legend, we think the Reviewer refers to the difference observed in Case 12-C2 and Case 13-C1. The main reason is the higher natural frequency of the cylinder, that is why the reduced velocity is lower than the additional cases. The increase in magnitude is observed between reduced velocity 5 and 10 on Case 13-C1, but the higher amplitudes remain in a similar range as the other cases.  A description with more details was added to the text.

“The results for most cases remain between reduced velocity 20 and 72, but Case 12-C2 and Case 13-C1 start in reduced velocity 6 due to the natural frequencies tested and it is higher than the additional cases tested. “ 

  1. Two peaks in spectra in Figure 4 are associated with the natural frequencies of the cylinders. The third one is explained by different reasons (“the vortex shedding”, “excitations due to the increase in the flow velocity”, “turbulence excitations”,…). Are these reasons really different? Or they are different descriptions of the same phenomenon? If the reasons are different, please, let us know how you detected them in different cases.

Thank you for the comment. The peaks related to natural frequencies observed can be related to the turbulence excitations, because of the spectrum of scales the response of the excitation remains around the natural frequencies.

The third peak observed around the Strouhal Number 0.2 in the cases cited and can be related to vortex shedding the excitation in this case is related to the periodic transversal force on the cylinder, although the excitations due to the increase in velocity and in turbulence contributes to levels of energy observed in the cases.  We add in the text additional information about these frequency peaks.

  1. The peak in Figure 4d at 85.9 Hz has very high value of the q-factor. Could it be a sign of an autogeneration process? Please, give your opinion about this result.

Thank you for your comment. In this case we at best could say that it is a self-excitation process that could happen by the vortex shedding for the first cylinder and the vibration movement that could feedback. In this case we do not identify this behavior due to the low amplitude observed, so the high q-factor must be related just to the low-rate energy loss.  

  1. Could you underline (for example, in the conclusion) the result, which are different from the previously published, and evaluate their importance for understanding of the studied problem?

Thank you for the comment. We added in the second paragraph of the conclusion some information about the importance and applications of the present study.

  1. It is necessary to give a brief analysis of the influence of obtained results on investigation in the field.

Thank you for the comment. We added in the introduction and in the second paragraph of the conclusion some information about the importance and applications of the present study.

  1. I think that the abstract may contain less details, but more conceptual points including reflections about two last remarks.

Thank you for the comment. We adjusted the abstract emphasizing the results and conclusions.

Minor:

  1. To make clearer the experiment description I recommend to add
  • the scheme of the cylinder’s attachment because Figure 2b is not clear;

Thank you for your comment. Figure 2b was changed to make clear the assembly used in the tests and included the axes, the direction measured by the accelerometer is y and this information was added in the text.

  • the sketch view of the cylinder’s oscillations corresponding to the first and second natural frequency;

The first frequency is related to movement of both blades for the same side and the second mode is related to the asymmetric movement of the blades. A schematic view of the modes is presented below.

We added this comment to the text.

  • direction of vibration measured by the accelerometer.

Thank you again. Figure 2b was changed to make clear the assembly used in the tests and included the axes, the direction measured by the accelerometer is y and this information was added in the text.

  • direction of vibration measured by the accelerometer.

Thank you again. Figure 2b was changed to make clear the assembly used in the tests and included the axes, the direction measured by the accelerometer is y and this information was added in the text.

  1. It would be useful to add clear definition of the mass ratio and the damping ratio. How did your adjusted them in the experiment?

Thank you for your comment. We added the description of mass ratio and damping ratio in the text:

“The mass ratio is relation between the cylinder mass and the mass of fluid displaced. The damping ratio is the relation between the energy dissipated in a cycle and the total structure energy in vibration. The parameter of mass ratio changed with the cylinder mass used in the assembly. The damping ratio changed with the length and thickness of the blades applied.  “

  1. Colored figures 4, 7, 10, 12 and 14 would be better for understanding. 

Thank you again. We adjusted the color pattern of the figures.

  1. The legend in Figure 3 seems to be chaotic. Please, rearrange names of curves. Also colored curves would be clearer.

Thanks for pointing this out; we agree that the legends were disorganized. We apologize for that. We organized the legend and used different colors for the cases with the first cylinder free to vibrate, the second cylinder free to vibrate, and both cylinders free to vibrate.

Please, see also attachment for a figure.

Reviewer 3 Report

1. Figure 14 should be colored edition.

Author Response

Thank you for the comment. We adjusted the figures color pattern.

Round 2

Reviewer 2 Report

Dear Authors, 

thank you for revision. Now the manuscript can be published.